# Facial Skin Lifting and Brightening Following Sleep on Copper Oxide Containing Pillowcases

**Gadi Borkow [1,*] and Adriana del Carmen Elías [2]**

[1]  Cupron Inc., 800 East Leigh Street, Suite 123, Richmond, VA 23219, USA
[2]  Biostatistics, Institute of Mathematics, Faculty of Biochemistry, Chemistry and Pharmacy (FBQyF), National University of Tucumán, San Miguel de Tucumán 4000, Argentina; eliasvacaflor@gmail.com
*  Correspondence: gadi@cupron.com; Tel.: +972-546-611-287

**Abstract:** Copper plays a key role in many of the physiological processes that occur in the skin. Previously it was found that sleeping on pillowcases impregnated with microscopic copper oxide particles results in reduction of wrinkles and fine lines. In the current study, it was examined if sleeping on copper oxide impregnated pillowcases results also in skin lifting and skin brightness. A four week, double blind, randomized study was performed, during which 45 women, aged 37–54, slept on copper oxide containing pillowcases (test group, $n = 23$) or on control pillowcases without copper oxide (control group, $n = 22$). Facial and eye skin surface was measured using an F-ray 3D measurement system and surface analysis was conducted using Image-pro® plus. Skin brightness was measured using a tristimulus colorimeter. Sleeping on the test pillowcases resulted in statistically significant skin lifting on the cheek area ($p = 0.039$) and eye area ($p = 0.001$) after four weeks of use as compared to baseline. The mean skin brightness in those sleeping on the test pillowcases increased after two ($p = 0.024$) and four weeks ($p = 0.008$). No statistically significant changes occurred during the study in the study participants using the control pillowcases. Statistically significant differences between both groups were recorded at two and four weeks for skin brightness and skin lifting, respectively. In conclusion, sleeping on copper oxide containing pillowcases results in facial skin lifting and brightness of the skin.

**Keywords:** copper oxide; pillowcases; skin; lifting; brightness; clinical study

## 1. Introduction

The dermal layer of the skin is composed mainly of fibroblasts and large flexible dynamic extracellular matrix (ECM) structures. These ECM structures, made up mostly of collagens I and III, elastin and fibrilin fibers, and glycosaminoglycan-rich proteoglycans, strongly interact with each other providing the skin with strength, extensibility and elasticity [1]. With age, the amount and size of fibroblasts in the dermis decreases, as does the production of ECM by fibroblasts that remain in the skin [2–6]. This results in changes of the skin's structure and appearance, including a marked loss of skin elasticity and recoil, increased appearance of wrinkles and facial sagging skin, especially under the eyes [7–11]. External influences, such as UV radiation, which may cause appearance of sunspots and uneven skin color, can also cause reduced skin elasticity, wrinkling and sagging [12,13].

Copper is an essential mineral involved in numerous human physiological processes [14]. In the skin, copper stimulates the production of collagens, integrin and fibronectin [15–18]. Copper upregulates lysyl oxidase, metalloproteinases, glycosaminoglycans and small proteoglycans involved in matrix remodeling, cell proliferation and re-epithelization [16,18–21]. Copper also stabilizes the extracellular matrix (ECM) once formed [18,22,23].

Adults should ingest 1–2 mg/day of copper as part of their normal diet. Copper is found in a wide variety of foods, including meat, seafood, nuts, seeds, avocado and chocolate, and in food multivitamins and additives. Trace amounts of copper ions are absorbed through the dermis when the skin is in contact with copper containing products [24,25]. Copper containing ointments are used, for example, in the treatment of cramps, swelling associated with trauma, rheumatic disease and disturbances of renal function [26]. Preparations containing copper in concentrations up to 20% were found not to cause any adverse reactions or toxicity [24].

A platform technology has been developed through which copper oxide particles are permanently embedded in textile fibers and fabrics [27,28]. Pillowcases, as well as socks, undergarments, linens, diapers, and a range of other consumer products embedded with copper oxide particles, are industrially produced and have been commercially available for many years now [29]. These products have been tested in numerous safety trials and found to be non-sensitizing and non-irritating [30,31]. Wearing socks impregnated with the copper oxide particles has been demonstrated to increase the skin elasticity in a double blind placebo controlled trial [32], and the use of such socks by diabetic individuals and soldiers has been hypothesized to protect their feet [33,34]. Similarly, in several clinical trials, it has been demonstrated that sleeping on pillowcases containing copper oxide particles reduces facial wrinkles and fine lines [35,36]. The current study, with the main aim to determine if sleeping on these pillowcases would result in reduced sagging, demonstrates that sleeping on these pillowcases for at least 5 h every night for a period of four weeks results in facial and eye skin lifting. Furthermore, this study demonstrates that sleeping on the pillowcases for two weeks also results in brightening of the facial skin.

## 2. Methods

### 2.1. Test Items

Two types of pillowcases made of 100% polyester yarns were tested. The test pillowcase (TP) contained 1% (weight/weight) copper oxide microscopic particles impregnated in the yarn (Figure 1), while the control pillowcase (CP) did not. The test and control items were differentiated by their color. The study participants and evaluators were blinded as to which pillowcase contained the copper oxide particles.

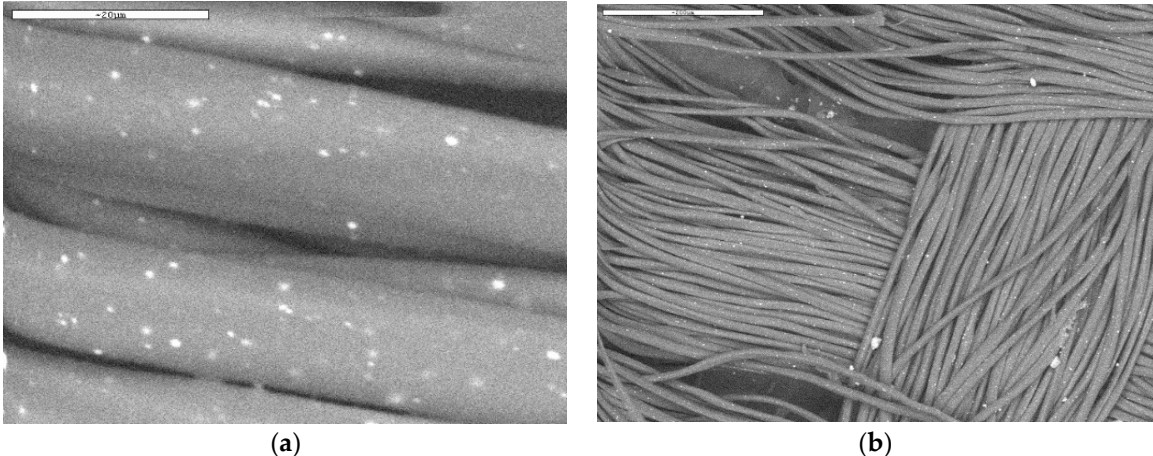

(**a**)          (**b**)

**Figure 1.** Scanning electronic microscope images (taken with a Jeol JMS 840 scanning electron microscope, Jeol, Tokyo, Japan) of the test pillowcase at two different magnifications: (**a**) 1500×; (**b**) 200×. The white dots are the copper oxide particles embedded in the polyester fibers.

## 2.2. Test Procedures

The study was conducted by the Dermapro Skin Research Center (Seoul, Korea), an institute specializing in the evaluation of safety and efficacy of cosmetics on human skin. The study was a double blinded, placebo controlled, randomized, parallel analytic case-control clinical trial. It was conducted during September and October of 2015 according to Good Clinical Practice (GCP) regulations and standard operating procedures of Dermapro. Appropriate study participants were selected according to the selection and exclusion criteria detailed in Table 1.

**Table 1.** Inclusion and exclusion criteria.

| Selection Criteria | Exclusion Criteria |
|---|---|
| 1. Women between 35 and 60 years of age<br>2. Presence of fine line wrinkles next to the eyes<br>3. Side sleepers<br>4. Agreed to use the pillowcases given to them throughout the 4 weeks of study<br>5. Volunteers should be cooperative and available during the study period<br>6. Signed a voluntary informed consent after the purpose and the protocol of the study were explained | 1. Pregnant or breastfeeding women<br>2. Women planning to become pregnant within the next 6 months<br>3. Women using or that have used during the last month products containing steroids for the treatment of skin diseases<br>4. Women with sensitive, irritable skin, acne, erythema and/or telangiectasia<br>5. Women treated with botox, other skin care products, or skin dermabrasion during the last 6 months<br>6. Women suffering from chronic diseases, such as asthma, diabetes, atopic dermatitis or hypertension<br>7. Women suffering from sleep disorders<br>8. Women who the main investigator felt would not comply with the test requirements<br>9. Participation in a previous study without an appropriate 3 month rest period in the interim |

The purpose of the study, the protocol, timetable, possible benefits, constraints linked to the study, and possible risks, were all explained to each study participant. After making a completely free decision, each participant willing to participate in the study signed an informed consent. The selected subjects were randomly divided into two groups: one group received the TP and one group received the CP. Each subject received two pillowcases, either the TP or the CP, and the same cleanser and moisturizer. The study participants were requested to solely use the supplied pillowcases every night during the whole study period as they would normally use their own pillowcase during their sleep. The use of other cosmetic products or skin pharmaceutical products during the study period was prohibited. The participants were asked to wash and dry the pillowcases as they would do with their regular pillowcases, but to avoid using textile softeners. The participants general skin facial characteristics, such as skin hydration, sebum content, and skin thickness, were evaluated by standard biophysical measurements performed routinely at Dermapro by using a Corneometer® CM 825 (Courage + Khazaka electronic GmbH, Cologne, Germany), a Sebumeter® SM 815 (Courage + Khazaka electronic GmbH) and ultrasonography with a Dermascan C® (Cortex Technology, Hadsund, Denmark), respectively. Skin sagging and brightness was determined before the commencement of the Study at Day 0, and after 2 and 4 weeks of using the pillowcases. The 4-week evaluation period was determined based on the previous studies [35,36] showing statistical significant reductions of facial wrinkles and fine lines at 4 weeks of using the pillowcases. All evaluations were performed under the same controlled environmental conditions (22 ± 2 °C) and relative humidity (50% ± 5%). Before the measurements, all participants removed their makeup, washed their faces and relaxed on a comfortable diagnostic bed under the controlled environmental conditions detailed above for at least 15 min.

## 2.3. Evaluation of Facial and Eye Skin Lifting

Facial lifting/sagging was measured using the F-ray three-dimensional (3D) surface measurement system (Beyoung, Seoul, Korea), which is based on moiré 3D topography analysis [37]. Briefly, the

study participants were placed in the F-ray 3D camera (Beyoung, Seoul, Korea) (Figure 2). The skin topography was determined by forming elongated circles by connecting a set of points based on absolute 3D coordinates (*x*, *y*, *z*) using determined control points (0, 0, 0) for each individual and photographs were taken. This was done at the onset of the trial (baseline, Day 0) and after 2 and 4 weeks for the cheek and eye skin areas (Figure 2). The skin lifting/sagginess was analyzed as described by Saito et al. [37] by using the Image-pro® plus software (Media Cybernetics, Rockville, MD, USA). By using the software program, skin movement along the *x*-, *y*- and *z*-axes can be measured and compared; for example, by analyzing the movement of the center area of the cheek, compared with the medial or lateral areas, and the lower area of the cheek compared with the upper area of the cheek [37].

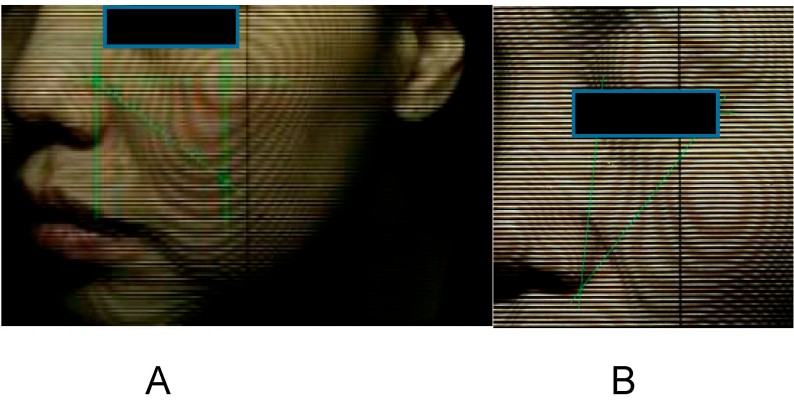

**Figure 2.** Skin contour lines of the cheek (**A**) and eye (**B**) areas.

### 2.4. Measurement of Skin Brightness

Skin brightness, (brightness factor or Luminance, L*, as defined by the International Commission on Illumination (CIE)), was measured 3 times at the same location on the cheek area with a tristimulus colorimeter (Spectrophotometer® CM-2500d, Minolta, Japan). The colorimeter measures the color spectral reflectance of the skin and gives three different values of the CIE color system—the brightness, color and chroma, which are represented as L*, a* and b*, respectively. The measurements were conducted at Day 0 before commencement of the study and after 2 and 4 weeks of using the pillowcases. For each determination, the average measured value was recorded.

### 2.5. Statistical Analysis

Wilcoxon signed-ranks tests, chi-square tests, Likelihood-ratio tests, paired *t*-tests and Repeated Measures ANOVA analyses were performed using the SPSS Package Program (version 20.0, IBM, Armonk, NY, USA). A statistically significant difference was set at $p \leqslant 0.05$. The increase or decrease of a given parameter was calculated according to the following equation: ((value at baseline − value after using the product for 2 or 4 weeks)/value at baseline)) × 100.

## 3. Results

### 3.1. General Characteristics of Study Participants

Forty-five women were recruited and divided into a Test Group (42–54 years old; mean age 47.23 ± 3.65; *n* = 22) and a Control Group (37–54 years old; mean age 44.96 ± 5.11; *n* = 23) following a computerized randomization. The sleeping habits and skin characteristics of the study participants compartmentalized by the study groups are detailed in Table 2. There were no statistically significant differences between both groups in terms of the general facial skin characteristics and sleeping habits at the onset of the study (Table 2).

**Table 2.** General and skin characteristics of the study participants.

| Parameter | Grade | Control Group Frequency (*n*) | Control Group Percentage (%) | Test Group Frequency (*n*) | Test Group Percentage (%) | *p* Value Chi-Square Test | *p* Value Likelihood Ratio Test |
|---|---|---|---|---|---|---|---|
| Skin Type | Dry | 8 | 34.78 | 11 | 50 | | |
| | Normal | 7 | 30.43 | 4 | 18.18 | | |
| | Oily | 2 | 8.70 | 0 | 0.00 | 0.341 | 0.248 |
| | Dry and oily | 6 | 26.09 | 7 | 31.82 | | |
| | Problematic | 0 | 0.00 | 0 | 0.00 | | |
| Hydration | Sufficient | 0 | 0.00 | 0 | 0.00 | | |
| | Normal | 11 | 47.83 | 8 | 36.36 | 0.436 | 0.436 |
| | Deficient | 12 | 52.17 | 14 | 63.64 | | |
| Sebum | Glossy | 2 | 8.70 | 4 | 18.18 | | |
| | Normal | 12 | 52.17 | 9 | 40.91 | 0.585 | 0.581 |
| | Deficient | 9 | 39.13 | 9 | 40.91 | | |
| Surface | Smooth | 6 | 26.09 | 2 | 9.09 | | |
| | Normal | 13 | 56.52 | 18 | 81.82 | 0.223 | 0.210 |
| | Rough | 4 | 17.39 | 2 | 9.09 | | |
| Thickness | Thin | 8 | 34.78 | 10 | 45.45 | | |
| | Normal | 14 | 60.87 | 10 | 45.45 | 0.178 | 0.169 |
| | Thick | 1 | 4.35 | 2 | 9.09 | | |
| Time of UV exposure per day | <1 h | 5 | 21.74 | 7 | 31.82 | | |
| | 1–3 h | 15 | 65.22 | 14 | 63.64 | 0.510 | 0.498 |
| | >3 h | 3 | 13.04 | 1 | 4.55 | | |
| Time of sleeping per night | <5 h | 1 | 4.35 | 0 | 0.00 | | |
| | 5–8 h | 22 | 95.65 | 21 | 95.45 | 0.368 | 0.250 |
| | >8 h | 0 | 0.00 | 1 | 4.55 | | |
| Sleeping habits | Front side | 0 | 0.00 | 0 | 0.00 | | |
| | Left side | 3 | 13.04 | 7 | 31.82 | 0.205 | 0.199 |
| | Right side | 5 | 21.74 | 6 | 27.27 | | |
| | Both sides | 15 | 65.22 | 9 | 40.91 | | |
| Irritability | Yes | 3 | 13.04 | 4 | 18.18 | 0.634 | 0.634 |
| | No | 20 | 86.96 | 18 | 81.82 | | |
| Stinging | Yes | 1 | 4.35 | 0 | 0.00 | 0.323 | 0.243 |
| | No | 22 | 95.65 | 22 | 100.00 | | |
| Smoking | Yes | 0 | 0.00 | 0 | 0.00 | 1.000 | 1.000 |
| | No | 23 | 100.00 | 22 | 100.00 | | |

### 3.2. Evaluation of Facial (Cheek) and Eye Skin Lifting

Compared to baseline (Week 0), there was a statistically significant increase in the mean cheek skin lifting (decreased sagging) after four weeks (*p* = 0.039) in those using the pillowcases containing copper oxide, whereas there was no statistically significant changes in the mean cheek skin lifting in those using the control pillowcases (Table 3, Figure 3). The differences between both groups reached statistical significance at four weeks (*p* = 0.021, Figure 3).

**Table 3.** Statistical analysis of facial skin lifting by contour line image analysis.

| Group | Week | Mean [1] | SD [2] | SEM [3] | *p*-Value [4] | Decrement (%) |
|---|---|---|---|---|---|---|
| Test Group | 0 | 37.79 | 5.09 | 1.09 | – | – |
| | 2 | 37.52 | 5.35 | 1.14 | 0.280 | 0.70 ▼ |
| | 4 | 37.20 | 4.78 | 1.02 | 0.039 * | 1.56 ▼ |
| Control Group | 0 | 37.95 | 4.64 | 0.97 | – | – |
| | 2 | 38.17 | 4.62 | 0.96 | 0.063 | 0.59 ▲ |
| | 4 | 38.51 | 4.89 | 1.02 | 0.097 | 1.48 ▲ |

[1] Decrement (▼) and increase (▲) of the mean value represents improvement or diminishment of skin lifting on cheek, respectively; [2] standard deviation; [3] standard error of the mean; [4] significantly different at * *p* < 0.05 compared with before using the pillowcase.

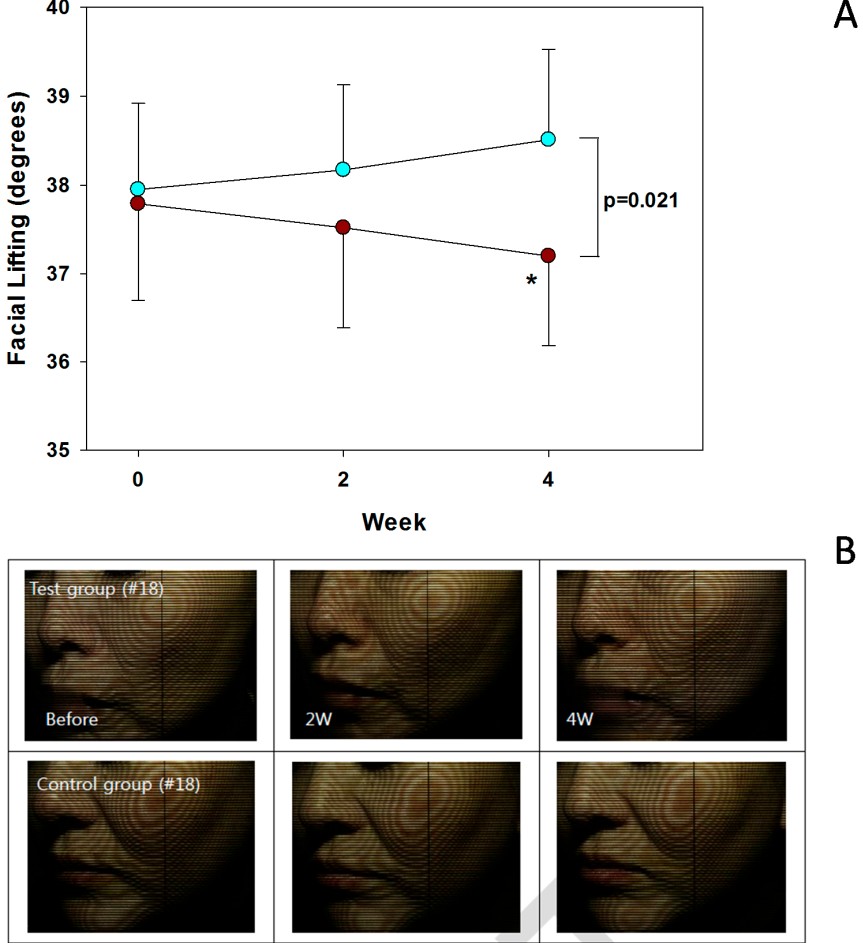

**Figure 3.** Comparison between the facial skin lifting (mean ± SEM) between the Control (⬤) and Test (⬤) Groups (**A**) and representative pictures (**B**). * *p* < 0.05 vs. before treatment.

Regarding the skin of the eye, there was a statistically significant decrease in the mean skin sagging after two (*p* = 0.011) and four weeks (*p* = 0.001) as compared to baseline (Week 0) only in the group of participants using the pillowcases containing copper oxide (Table 4, Figure 4). No statistically significant changes occurred on the mean eye skin lifting in those using the control pillowcases (Table 4, Figure 4). The differences between both groups reached statistical significance at four weeks (*p* = 0.003, Figure 4).

**Table 4.** Statistical analysis of eye skin lifting by contour line image analysis.

| Group | Week | Mean [1] | SD [2] | SEM [3] | *p*-Value [4] | Decrement (%) |
|-------|------|----------|--------|---------|---------------|---------------|
| Test Group | 0 | 91.43 | 3.04 | 0.73 | – | – |
| | 2 | 91.86 | 3.19 | 0.68 | 0.011 * | 0.46 ▲ |
| | 4 | 92.01 | 3.29 | 0.70 | 0.001 * | 0.63 ▲ |
| Control Group | 0 | 91.74 | 3.84 | 0.80 | – | – |
| | 2 | 91.76 | 3.88 | 0.81 | 0.881 | 0.02 ▲ |
| | 4 | 91.68 | 3.89 | 0.81 | 0.505 | 0.07 ▼ |

[1] Increment (▲) and decrease (▼) of the mean value represents improvement or diminishment of skin lifting on the eye area, respectively; [2] standard deviation; [3] standard error of the mean; [4] significantly different at * *p* < 0.05 compared with before using the pillowcase.

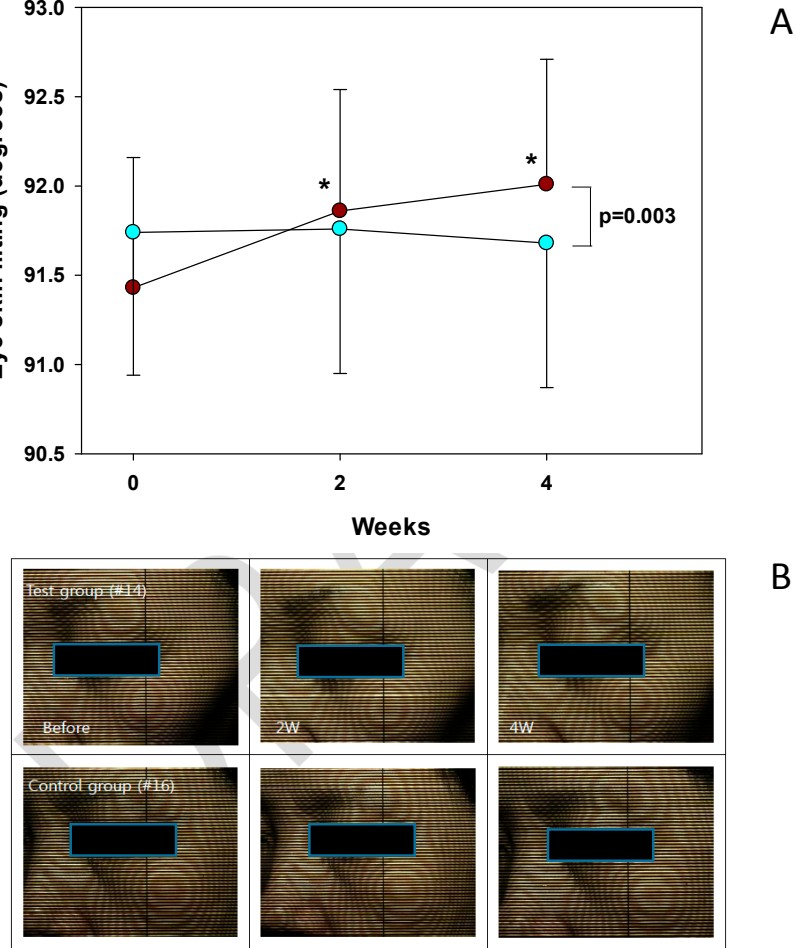

**Figure 4.** Comparison between the eye skin lifting (mean ± SEM) between the Control (⬤) and Test (⬤) Groups (**A**) and representative pictures (**B**). * $p < 0.05$ vs. before treatment.

### 3.3. Evaluation of Skin Brightness

Compared to baseline (Week 0), there was a statistically significant increase in the mean skin brightness after two ($p = 0.024$) and four weeks ($p = 0.008$) in those using the pillowcases containing copper oxide, whereas there was no statistically significant changes in the skin brightness in those using the control pillowcases (Table 5 and Figure 5). The differences between both groups was statistically significantly different at two ($p = 0.002$) and four weeks ($p = 0.002$) (Figure 5).

**Table 5.** Statistical analysis of skin brightness (L* value) by spectral reflectance.

| Group | Week | Mean [1] | SD [2] | SEM [3] | *p*-Value [4] | Decrement (%) |
|---|---|---|---|---|---|---|
| | 0 | 64.15 | 2.30 | 0.49 | – | – |
| Test Group | 2 | 64.38 | 2.20 | 0.47 | 0.024 * | 0.37 ▲ |
| | 4 | 64.44 | 2.26 | 0.48 | 0.008 * | 0.45 ▲ |
| | 0 | 64.01 | 1.95 | 0.41 | – | – |
| Control Group | 2 | 63.91 | 1.94 | 0.40 | 0.108 | 0.15 ▼ |
| | 4 | 63.93 | 1.84 | 0.38 | 0.139 | 0.13 ▼ |

[1] Increment (▲) and decrease (▼) of the L* value represents improvement or diminishment of skin brightness, respectively; [2] standard deviation; [3] standard error of the mean; [4] significantly different at * $p < 0.05$ compared with before using the pillowcase.

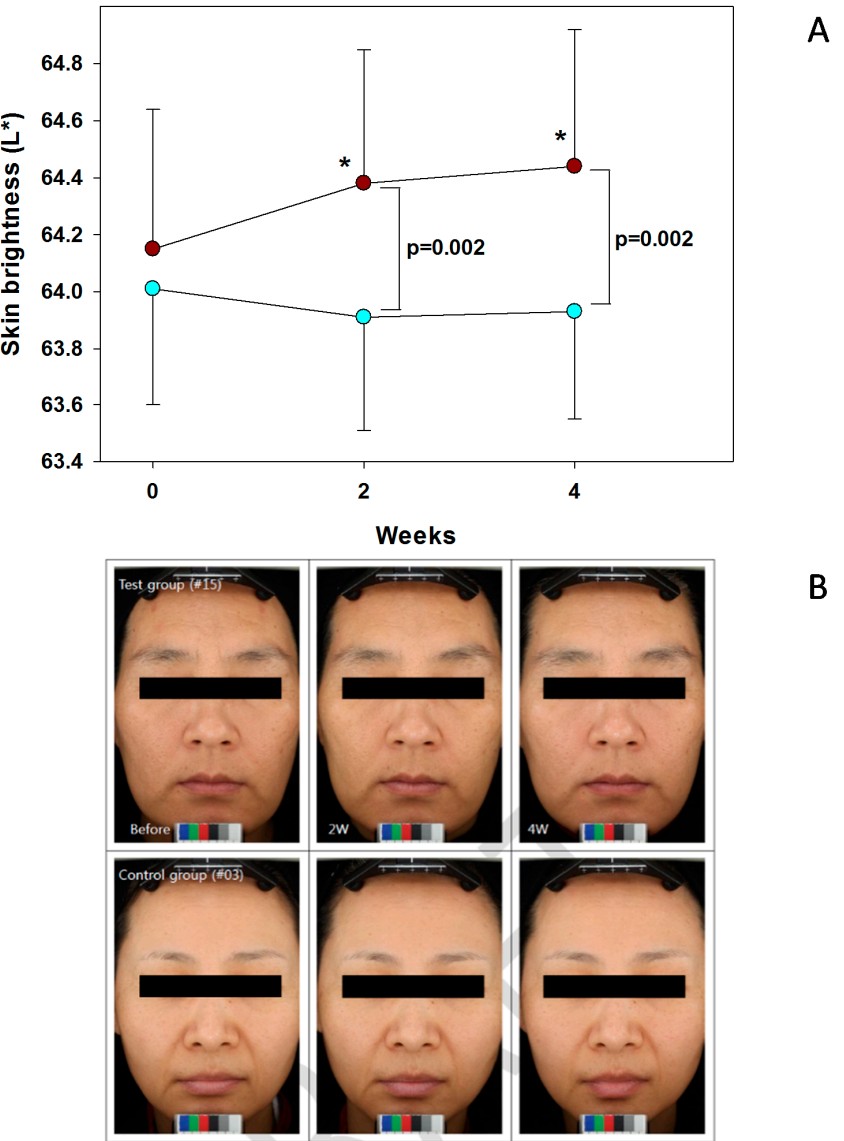

**Figure 5.** Comparison between the facial skin brightness (mean ± SEM) between the Control (○) and Test (●) Groups (**A**) and representative pictures (**B**). * *p* < 0.05 vs. before treatment.

### 3.4. Adverse Reactions

No adverse reactions defined as itching, prickling, burning, stinging, tightness, burning of the eyes, weeping, erythema, edema, scaling, papule or any other reactions were noted in all 45 study participants both after two and four weeks of using the control or test pillowcases.

## 4. Discussion

As clearly demonstrated in this placebo-controlled trial, sleeping on copper oxide containing pillowcases for a minimum of 5 h per night results in facial skin lifting. We postulate the following model to explain the obtained results (Figure 6): (a) the copper oxide particles embedded in the pillowcase fibers serve a "slow release" reservoir of copper ions that are released at ppm levels into the moisture (data not shown) found between the face and the pillowcase; (b) the copper ions are absorbed through the skin. The capacity of copper ions to be absorbed through the skin was already demonstrated [24,25]; (c) once absorbed, the copper ions induce proliferation of fibroblast cells; stimulation of dermal fibroblasts proliferation by copper ions has been demonstrated [38];

(d) the absorbed copper ions also stimulate the production and secretion of collagen, fibronectin and integrin by the dermal fibroblasts, as previously demonstrated [15,39]; (e) new ECMs are formed, and following the ECM formation, the copper ions stabilize the ECMs directly [22,40] and indirectly via serving as a cofactor of several enzymes, such as lysyl oxidase, needed for efficient ECM protein cross-linking [16]; (f) the newly formed and stabilized ECMs increase the firmness of the facial skin and its elasticity [1]. Copper oxide containing socks have been shown to increase the skin elasticity upon use [32]; and (g) increased skin elasticity and firmness results in reduced sagginess [41].

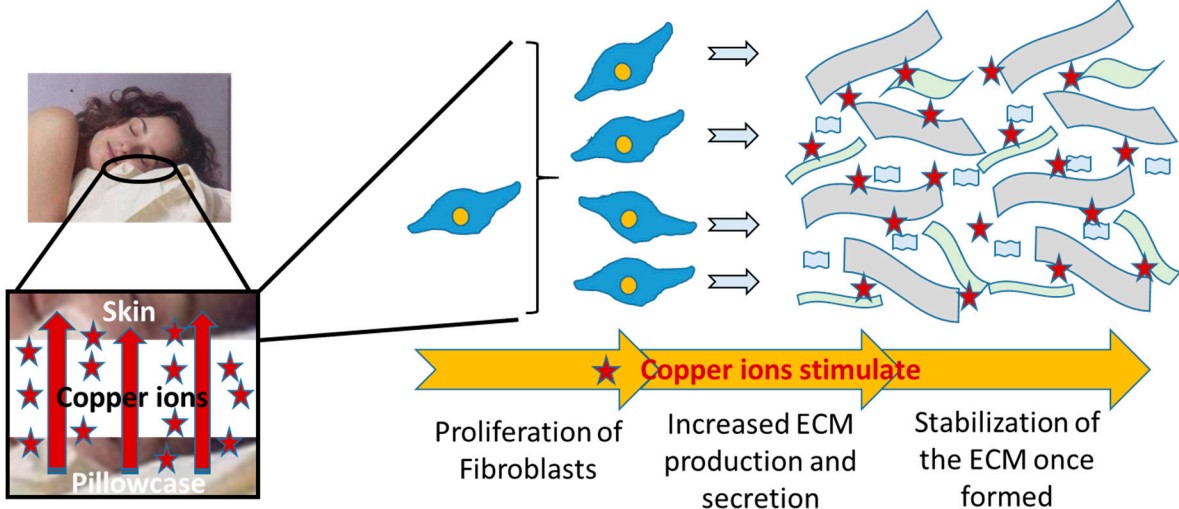

**Figure 6.** Model explaining facial skin lifting by copper oxide containing pillowcases.

Another clear result of this study was the increased brightness that occurred in the study participants using the copper oxide containing pillowcases. In the past, we observed that individuals using gloves containing copper oxide particles had reduced hyperpigmentation already after two weeks of using the gloves (unpublished data). Copper has been shown to modulate integrins expression by keratinocytes [42], the cells that constitute ~90% of the epidermis. We hypothesize that copper affects the keratinocytes in other ways too that may result in lighter skin. Obviously this observation (brightening of the skin) and our hypothesis require further study and substantiation. Copper also attaches itself to the enzyme superoxide dismutase, an enzyme important in protection against free radicals that is present in the skin [43].

Sleeping habits may influence how the pillowcase may affect the facial skin, as some individuals sleep more on one side than the other. During sleep, individuals do change positions and sides, but overall some skin areas may be more in contact with the pillowcase than other parts and thus be affected more than the areas not in contact with the pillowcase. This study did not analyze the effect of the sleeping habits. In addition, this study did not study skin changes following cessation of exposure to the pillowcase. Finally, this study only examined the effects on the skin for a period of one month. The effect for longer periods, the effect of sleeping habits and what happens after cessation of use are questions that should be examined in subsequent studies.

This study, showing apparent reduced sagginess via affecting the skin ECM, is in accordance with previous studies with copper oxide embedded pillowcases, which showed that sleeping on them results in reduction of fine lines and wrinkles [35,36]. Both processes, sagging and wrinkle appearance, result from changes in the skin ECM, and as demonstrated, copper ions released from copper oxide particles, enhance the production of ECM proteins by dermal fibroblasts [15]. In addition, similar to thirteen previous studies with pillowcases and other consumer products containing copper oxide [30,31,35,36,44,45], the use of copper oxide containing pillowcases did not cause skin irritation, itching, or any other adverse reactions. The amount of copper ions released to the skin moisture is

in the ppm range (unpublished data). Copper oxide impregnated products such as socks, apparel, adult diapers, and hospital linens, are in use worldwide for several years, with no reports of adverse reactions. This is in accordance with the very low risk of adverse reactions due to dermal copper contact [46]. Copper oxide is found in multivitamin pills and dietary supplements since copper is an essential mineral and a daily oral consumption of 1–2 mg is recommended [14].

## 5. Conclusions

This study demonstrates that sleeping on fabrics that liberate copper ions not only reduces wrinkles as previously demonstrated, but also reduces skin sagging. Sleeping on copper oxide impregnated pillowcases is thus a safe and effective way to help maintain the skin firmness and young appearance.

**Acknowledgments:** This study was funded by Two and Up C. Ltd., Seoul, Korea.

**Author Contributions:** Gadi Borkow help design the study and wrote the manuscript. Adriana del Carmen Elías conducted the statistical analysis.

**Conflicts of Interest:** Gadi Borkow is the Chief Medical Scientist of Cupron Inc. Cupron Inc. developed the technology of embedding copper oxide particles in textiles. Adriana del Carmen Elías has no conflict of interests.

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
