# Peer review of "Facial Skin Lifting and Brightening Following Sleep on Copper Oxide Containing Pillowcases"

_cosmetics, doi:10.3390/cosmetics3030024_

Reviewer 1 Report

The article “Facial Skin Lifting and Brightening following Sleep 2 on Copper Oxide Containing Pillowcases” demonstrated the beneficial effects of copper oxide particles on the skin.

I found the manuscript very interesting, it is well written and the data presented are exhaustive and in agreement with the conclusions that the authors presented. The technology presented was also innovative and intriguing.

Although one important issue was not addressed at all by the authors, which was the proved cytotoxicity of copper oxide particles on the skin. Recent papers clearly demonstrated that copper particles, above a certain concentration, may have harmful effect on skin cells, both fibroblasts and keratinocytes. They may also activate apoptotic mechanisms and cause DNA damages (Luo et al, 2014, Int J Nanomed; Alarifi et al 2013, Int J toxicol).

In this article only the beneficial effect of Cu was described but the potential toxicity of the Cu particles never mentioned, especially if Cu was used above a certain concentration.

Could the authors address this point? Do the authors have any idea on the amount of copper particles that is delivered on the skin by the pillow cases?  Could they talk about the limit of their technology in order to keep under control the safety issue?  Please add this info in the introduction and/or discussion part.

Author Response

Dear editor,

I would like to thank you and the reviewers for helping us improve our manuscript entitled “Facial Skin Lifting and Brightening following Sleep on Copper Oxide Containing Pillowcases”.

We have carefully revised the manuscript according to the Reviewers comments. The revisions in the manuscript are highlighted in red. Below are the responses, point by point, to the Reviewer comments. For ease, the Reviewer comments are in black and our responses are in blue.

We hope very much that you will now find the manuscript appropriate for publication in Cosmetics.

Sincerely,

Dr. Gadi Borkow and Dr. Adriana del Carmen Elías

Reviewer 1:

The article “Facial Skin Lifting and Brightening following Sleep 2 on Copper Oxide Containing Pillowcases” demonstrated the beneficial effects of copper oxide particles on the skin.

I found the manuscript very interesting, it is well written and the data presented are exhaustive and in agreement with the conclusions that the authors presented. The technology presented was also innovative and intriguing.

Although one important issue was not addressed at all by the authors, which was the proved cytotoxicity of copper oxide particles on the skin. Recent papers clearly demonstrated that copper particles, above a certain concentration, may have harmful effect on skin cells, both fibroblasts and keratinocytes. They may also activate apoptotic mechanisms and cause DNA damages (Luo et al, 2014, Int J Nanomed; Alarifi et al 2013, Int J toxico).

In this article only the beneficial effect of Cu was described but the potential toxicity of the Cu particles never mentioned, especially if Cu was used above a certain concentration.

Could the authors address this point? Do the authors have any idea on the amount of copper particles that is delivered on the skin by the pillow cases?  Could they talk about the limit of their technology in order to keep under control the safety issue?  Please add this info in the introduction and/or discussion part.

The safety of using the pillowcases is obviously a very important issue. It should be noted that the copper oxide particles embedded in the studied pillowcases are not nanoparticles. Their size is at least 10 times the size of the nanoparticles studied in the two manuscripts mentioned above (Luo et al, 2014, Int J Nanomed; Alarifi et al 2013, Int J toxicol). Nanoparticles below 100 nm of size demonstrate significant more cytotoxicity than particles that are above 100 nm. In our case, the size of the particles ranges between 500 to 2000 nm.  Be as it may, also copper oxide particles may be cytotoxic, as cells in cell culture may be more susceptible to the environment than when in tissue such as when being part of the skin. Copper oxide is basically non-soluble, and the amount of copper ions that are liberated into the skin moisture are ppm (less than 0.5 nM), concentrations that do not cause cytotoxicity, as we reported previously (Philips et al. Beneficial regulation of fibrillar collagens, heat shock protein-47, elastin fiber components, transforming growth factor-beta1, vascular endothelial growth factor and oxidative stress effects by copper in dermal fibroblasts. Connect Tissue Res 2012, 53, 373-378). No apoptosis, DNA damage, or any atypia was noted. On the contrary, the presence of copper oxide at 0.5 nM, in addition to stimulating the production of extracellular skin proteins (as discussed in the Discussion section), importantly confer protection to the dermal cells from oxidative stress. However, this was demonstrated in cell culture and not in whole skin tissue or skin biopsies, and thus mentioning this in the manuscript seems to us misleading.

As mentioned in the Results section, no adverse reactions were noted in any of the study participants. Similarly, in 13 previous clinical studies with different textiles as well as pillowcases impregnated with the copper oxide particles not a single adverse reaction was recorded. Thousands of consumer products containing copper oxide particles, including pillowcases, are being sold yearly, with no reports of any adverse effects. Medical textiles that contain copper oxide particles are in use more than 3 years now, with more than 150,000 patient days, with not even one adverse reaction recorded. The safety of the copper oxide impregnated products was mentioned in the discussion section in the following paragraph to which the underlined parts were now added (lines 224-232): “Also, in similarity to previous thirteen studies with pillowcases, hospital linens and other consumer products containing copper oxide [30,31,35,36,44,45], the use of copper oxide containing pillowcases did not cause skin irritation, itching, or any other adverse reactions. The amount of copper ions released to the skin moisture is in the ppm range (unpublished data). Copper oxide impregnated products such as socks, apparel, adult diapers, and hospital linens, are in use worldwide for several years, with no reports of adverse reactions. This is in accordance with the very low risk of adverse reactions due to dermal copper contact [46]. Copper oxide is found in multivitamin pills and dietary supplements since copper is an essential mineral and an oral daily consumption of 1-2 mg is recommended [14].”  

Reviewer 2 Report

Overall, this is a well written, albeit limited, study on a novel topic. 

For the reader not familiar with the concepts of measuring skin tightening and brightening, a brief review of the F Ray 3D measurement  and the Image pro plus systems would be helpful.  How do statistically significant changes correlate with clinically evident findings?

The authors should address how sleeping habits may impact the results, ie individuals often predominantly sleep on one side of their face.  How was this accounted for in the analysis of the results?

A few additional questions for the authors to consider addressing: How was the length of the study chosen? How would the duration of exposure be expected to impact the effects? What is the natural course of the skin changes following cessation of exposure?

Author Response

Dear editor,

I would like to thank you and the reviewers for helping us improve our manuscript entitled “Facial Skin Lifting and Brightening following Sleep on Copper Oxide Containing Pillowcases”.

We have carefully revised the manuscript according to the Reviewers comments. The revisions in the manuscript are highlighted in red. Below are the responses, point by point, to the Reviewer comments. For ease, the Reviewer comments are in black and our responses are in blue.

We hope very much that you will now find the manuscript appropriate for publication in Cosmetics.

Sincerely,

Dr. Gadi Borkow and Dr. Adriana del Carmen Elías

Reviewer 2:

Overall, this is a well written, albeit limited, study on a novel topic.

For the reader not familiar with the concepts of measuring skin tightening and brightening, a brief review of the F Ray 3D measurement  and the Image pro plus systems would be helpful.  How do statistically significant changes correlate with clinically evident findings?

In the original manuscript, the skin tightening measurement were described and the manuscript describing very detailed the methodology was given (now, in the revised manuscript, lines 106 to 113). In the revised manuscript another sentence (lines 113-116) was added to help the explain how the skin movement over time (sagging) is measured. In addition, a sentence further explaining the measurement of skin brightness was added to the revised manuscript (lines 122-124). 

The authors should address how sleeping habits may impact the results, ie individuals often predominantly sleep on one side of their face.  How was this accounted for in the analysis of the results? 

The following new sentences were added to the Discussion: “Sleeping habits may influence how the pillowcase may affect the facial skin, as some individuals sleep more on one side than the other. During sleep individuals do change positions and sides, but overall some skin areas may be more in contact with the pillowcase than other parts and thus be affected more than the areas not in contact with the pillowcase. This study did not analyze the effect of the sleeping habits.”. (lines 212-216).

A few additional questions for the authors to consider addressing: How was the length of the study chosen?

The length of the study was chosen based on the previous referenced studies with the pillowcases that found reduction of facial wrinkles and fine lines already after 2 and 4 weeks of using the pillowcases.  This is now mentioned in lines 99-101 of the revised manuscript. 

How would the duration of exposure be expected to impact the effects? What is the natural course of the skin changes following cessation of exposure?

The following sentences were added to the Discussion section in reference to the above mentioned issues: “In addition, this study did not study skin changes following cessation of exposure to the pillowcase. Finally, this study only examined the effects on the skin for a period of one month. What would be the effect for longer periods, the effect of sleeping habits and what happens after cessation of use, are questions that should be examined in subsequent studies.”. (lines 216-219)

Reviewer 3 Report

line 91: Please clarify the sleep behavior. What about study participants who are consequently back sleeper. Where htere any standardization?

line 96 /97: Where participants acclimatzed?

line 128: how did you collect the skin characteristics like hydration, sebum surface and thickness? Did you used standardized biophysical measurements?

line 177: why does skin lifting effects and skin brightness increase after using a pillow free of copper?

line 180: how did you verify that the face stayed in contact with the pillow over an period of 5 hours? Could it be possible that some participants sleeped on onla one side of the face over the study period?

Author Response

Dear editor,

I would like to thank you and the reviewers for helping us improve our manuscript entitled “Facial Skin Lifting and Brightening following Sleep on Copper Oxide Containing Pillowcases”.

We have carefully revised the manuscript according to the Reviewers comments. The revisions in the manuscript are highlighted in red. Below are the responses, point by point, to the Reviewer comments. For ease, the Reviewer comments are in black and our responses are in blue.

We hope very much that you will now find the manuscript appropriate for publication in Cosmetics.

Sincerely,

Dr. Gadi Borkow and Dr. Adriana del Carmen Elías

Reviewer 3:

line 91: Please clarify the sleep behavior. What about study participants who are consequently back sleeper. Where there any standardization?

As specified in Table 1, only side sleepers were included in the study. No back sleepers were included. Obviously, during the night it may be that some of the side sleepers sleep on their back for a given period of the night. We could not control for that, as participants slept in their house without being monitored. When comparing the sleeping habits of the participants in both studied group (as specified in Table 2), there were no statistical significant differences. 

The following new sentences were added to the Discussion: “Sleeping habits may influence how the pillowcase may affect the facial skin, as some individuals sleep more on one side than the other. During sleep individuals do change positions and sides, but overall some skin areas may be more in contact with the pillowcase than other parts and thus be affected more than the areas not in contact with the pillowcase. This study did not analyze the effect of the sleeping habits.”. (lines 212-219).

line 96 /97: Where participants acclimatized?

The “standard of test” in Dermapro is to do all measurement under a controlled environment after the study participants were acclimatized. This is now specified in the revised manuscript, line 104.

line 128: how did you collect the skin characteristics like hydration, sebum surface and thickness? Did you used standardized biophysical measurements?

These are standard biophysical measurements performed at Dermapro Center (http://www.dermapro.co.kr/eng/service/efficacy_face.php). Specifically, hydration was evaluated with a Corneometer(®) CM 825, sebum with a Sebumeter(®) SM 815, and skin thickness by ultrasonography by using a Dermascan C(®). This is now specified in the revised manuscript, lines 94-97.

line 177: why does skin lifting effects and skin brightness increase after using a pillow free of copper

In line 177 of the original manuscript it is not specified that skin lifting or skin brightness increases in pillowcases without copper. On the contrary, it is discussed why it increases in pillowcases with copper. As mentioned in the Results section, and detailed in Tables 3, 4 and 5, in the control group using pillowcases without copper there were no statistically significant changes in the skin lifting or brightness, but only in the group using the pillowcases with copper.

line 180: how did you verify that the face stayed in contact with the pillow over an period of 5 hours? Could it be possible that some participants sleeped on onla one side of the face over the study period? 

We could not verify that the face stayed in contact with the pillowcases over a period of 5 hours. The study participants were asked to use the test pillowcases as they are used to use their regular pillowcases. It may be that some of the participants in both groups slept only on one side of the face. However, even though some people are right or left sleepers, most people, including right and left sleepers, switch sides a few times per night while they sleep.

As mentioned above, the following new sentences were added to the Discussion: “Sleeping habits may influence how the pillowcase may affect the facial skin, as some individuals sleep more on one side than the other. During sleep individuals do change positions and sides, but overall some skin areas may be more in contact with the pillowcase than other parts and thus be affected more than the areas not in contact with the pillowcase. This study did not analyze the effect of the sleeping habits.”. (lines 212-216).

Round  2

Reviewer 1 Report

The manuscript has improved and the recommendations were addressed. It is fine to be published in the present form.